# Complete Uterine Septum, Double Cervix and Vaginal Septum (U2b C2 V1): Hysteroscopic Management and Fertility Outcomes—A Systematic Review

**DOI:** 10.3390/jcm12010189

**Published:** 2022-12-26

**Authors:** Luca Parodi, Ilda Hoxhaj, Giorgia Dinoi, Mariateresa Mirandola, Federica Pozzati, Ghergana Topouzova, Antonia Carla Testa, Giovanni Scambia, Ursula Catena

**Affiliations:** 1Faculty of Medicine and Surgery, Università Cattolica del Sacro Cuore, 00168 Rome, Italy; 2Obstetrics and Gynecology Department, Lavagna Hospital, ASL4 Liguria, 16033 Lavagna, Italy; 3Department of Surgical Oncology and Gastroenterology, Università di Padova, 35128 Padova, Italy; 4Dipartimento della Salute della Donna, del Bambino e di Sanità Pubblica, Fondazione Policlinico Universitario A. Gemelli IRCCS, 00168 Rome, Italy

**Keywords:** U2bC2V1, Müllerian malformation, uterine malformation, hysteroscopy, metroplasty, infertility

## Abstract

Background: complete uterine septum, double cervix and vaginal septum is a rare complex Müllerian anomaly affecting patients’ quality of life in terms of fertility and pelvic pain. The aim of our review is to gather the studies concerning the diagnosis and treatment this complex malformation and to describe the related fertility outcomes. Methods: this study was conducted in 2022, according to the criteria of Preferred Reporting Items for Systematic Reviews and Meta-analyses (PRISMA) and the protocol was submitted to the International Prospective Register for Systematic Reviews (PROSPERO). PubMed, Scopus and Web of Science electronic databases were searched to find eligible articles. In total, 538 articles were identified through literature research. A total of ten articles satisfied the eligibility criteria and were included in the systematic review. Results: 86 affected women were evaluated, and 71 of them were treated. Almost all patients included in our research presented with primary infertility or with a history of recurrent miscarriages; half of all patients also reported dyspareunia. After surgical treatment, 47 pregnancies were achieved: 41 live birth and ongoing pregnancies and six spontaneous miscarriages occurred; a significantly lower miscarriage rate was reported after surgical treatment. Conclusion: hysteroscopic treatment of U2b C2 V1 anomaly can be safely performed, leading to favorable fertility outcomes, measured as the achievement of pregnancy and a reduction in miscarriage rate.

## 1. Introduction

Complete uterine septum, double cervix and vaginal septum is a rare complex Müllerian malformation defined as Class U2b C2 V1 by the ESHRE/ESGE classification 2013 [1]. The first studies concerning female genital tract embryogenesis hypothesized that the uterus forms after fusion of the two Müllerian ducts and subsequent septal reabsorption which proceeds unidirectionally, from the caudal to the cephalic direction. The Class U2b C2 V1 malformation is not explained by this traditional embryologic Müllerian theory. Indeed, more recently, the bidirectional reabsorption theory has been widely recognized. According to this theory, septal reabsorption starts at the level of the isthmus proceeding simultaneously in a cranial and caudal direction. This theory was first described in the literature by McBean et al. in 1994 [2]. Uterine septum alone is considered one of the most common uterine anomalies and represents 35% of cases, with a prevalence of 1% to 2% in the general population. Uterine septum has been associated with serious reproductive implications, such as infertility, pregnancy loss and obstetrical complications such as preterm deliveries and intrauterine growth restriction [3,4]. According to the literature, abnormal blood supply to the septum leads to spontaneous miscarriages, indicating that normal uterine cavity restoration could be the best therapeutic approach [5]. In rare cases, uterine septum is associated with cervical duplication and longitudinal vaginal septum. Vaginal longitudinal non-obstructive septum is frequently associated with dyspareunia. The lack of standardized diagnostic techniques and the low incidence of this malformation in the general population may lead to frequent misdiagnosis. The more recently defined role of 3D-ultrasound [6] in combination with diagnostic hysteroscopy has improved the fine diagnosis of complex Müllerian malformations, including Class U2b C2 V1, leading to newer and more detailed classifications [1,7]. Despite the improvement achieved, the true incidence of this triad (uterine septum with cervical duplication and longitudinal vaginal septum) is still unknown. Moreover, in these patients, surgical uterine septum resection techniques are not standardized and the role of surgery is still not clear. The National Institute for Health and Care Excellence (NICE) [8] and American Society of Reproductive Medicine (ASRM) [9] support hysteroscopic septum resection, whereas the European Society of Human Reproduction and Embryology (ESHRE) and the Royal College of Obstetricians and Gynaecologists (RCOG) [10] consider that there is not enough evidence. There are few studies regarding Class U2b C2 V1 malformations, mostly small case series and retrospective papers, describing diagnostic techniques, surgical treatments, and obstetrical outcomes. The aim of our review is to assemble and summarize all the studies present in the literature concerning the diagnosis and treatment of this rare and complex Müllerian malformation.

## 2. Materials and Methods

This study was conducted according to the criteria of Preferred Reporting Items for Systematic Reviews and Meta-analyses (PRISMA) [11] and the protocol has been submitted to the International Prospective Register for Systematic Reviews (PROSPERO) [12].

### 2.1. Literature Research

PubMed, Scopus and Web of Science electronic databases were searched to find eligible articles published in English, from 2000 to May 2022. No additional filters were applied to the search strategy. The research was started in February 2022 and completed in May 2022. A combination of search and MeSH terms were employed in the search query: “uterine septum”, “septate uterus”, “double cervix”, “vaginal septum”, “hysteroscopy treatment”, “hysteroscopic correction”, “metroplasty”, “septal resection”, “reproductive outcome”, and “infertility”. Studies evaluating infertile women with U2b C2 V1 malformation and providing reproductive outcomes such as pregnancy rate, miscarriage rate and/or obstetrical outcomes (live birth rate, malpresentation, preterm delivery) after the treatment of this complex anomaly were considered eligible. We included randomized controlled trials, observational studies, case reports or case series in the systematic review, whereas editorials, comments, video articles, conference papers and review papers were not considered. We excluded publications in which the cervical assessment was not clear: we included patients with double cervix and we excluded patients with single cervix and cervical septum. Moreover, we excluded publications concerning other types of malformations, articles whose full text was not available or studies that did not explain the type of treatment adopted completely and clearly. Two independent researchers (LP, GD) screened the retrieved results by title and abstract, and in a second step carefully reviewed articles with the full text available. The RYYAN application [13] was used to complete the review research by both authors permitting a blind and masked screening of references. Studies satisfying the eligibility criteria were selected for inclusion. Any discrepancy was resolved through discussion. Study screening and selection process was reported in a PRISMA flow chart (Figure 1). For each of the included studies, we extracted the following data: title, first author, publication year, country, type of disease, population size, treatment and reported outcomes. 

### 2.2. Quality Assessment

The quality assessment of the included studies was conducted using the Checklist from Joanna Briggs Institute [14], according to the study design: Checklist for case-control studies, Checklist for case series and Checklist for case-reports. The Checklist for case series and the Checklist for case reports evaluated case definition, selection, description, risk of bias, treatment, adequate reporting, and statistical analysis, indicating that an answer of ‘no’ to any of the questions negatively impacts the quality of the case series. The Checklist for case control studies evaluated whether the groups were comparable or matched appropriately, if the exposure and the outcomes were measured in a standard, valid and reliable way, for both cases and controls, if confounding factors were identified and if the statistical analysis was correct. In all the case-reports patients’ demographic characteristics, history, clinical conditions, assessment methods and treatment procedures were clearly described, whereas adverse effects related to the treatment option were reported in only one study. Case-series did not clearly report demographic data and did not indicate whether the condition of interest was measured in a standard, reliable way for all the participants. In the case-control studies, case and controls were comparable and matched appropriately; outcomes were measured in a standard, valid and reliable way, and the statistical analysis was appropriate, whereas they did not report any data regarding exposure measurement and confounding factors. 

## 3. Results

In total, 538 articles were identified through literature research. After the removal of duplicates (n = 202), 336 studies were included initially. Subsequently, editorials, comments, video articles, conference papers and review papers were excluded (n = 315), leaving 21 articles. Four studies were excluded from this final pool because they did not concern the U2b C2 V1 malformation, two because they were abstracts, three because the full text was not available and two because surgical treatment adopted was not clearly described. A total of ten articles satisfied the eligibility criteria and were included in the systematic review (Figure 1). Given the limited quantitative data, it was not possible to carry out a meta-analysis. Characteristics of the included studies are reported in Table 1. They were published between 2004 and 2020, and were conducted in Turkey, Slovenia, China, Germany, United States, Japan and the United Kingdom (UK). For the study design, we identified only case reports/series, one retrospective case-control study, one retrospective case-series study and one prospective observational study. Overall, 86 women with the U2b C2 V1 malformation with a median age of 25 years old were evaluated, and 71 of them were treated. Almost all patients included in our research presented with primary infertility or with a history of recurrent miscarriages; half of all patients also reported dyspareunia.

### 3.1. Diagnosis and Classification

The classification system used for the definition of the malformation varied among the studies. Most of the included articles used the American Fertility Society (AFS) classification [15,17,22,23] and one article adopted the Vagina Cervix Uterus Adnex-associated Malformation (VCUAM) classification [16]. One article adopted the ESHRE/ESGE terminology considering that most of the studies were published before the updated ESHRE/ESGE nomenclature in 2013 [24]. The diagnosis of uterine septum differed between studies as follows: three-dimensional ultrasound (3DUS) in one study, a combination of Magnetic Resonance Imaging (MRI) and trans-vaginal ultrasound (TVUS) in three studies, hysterosalpingography (HSG) in one study, and MRI, TVUS and HSG together in three studies. Hysteroscopy was integrated as a diagnostic tool in two studies. 

### 3.2. Surgical Technique

Surgical techniques used also differed. Vaginal septum was mainly resected using Metzenbaum cold scissors; the surgical removal of the complete uterine septum (metroplasty) was mostly performed through hysteroscopy, except in one study published in 2004, where five cases were treated with a modified Tompkins laparotomic metroplasty [15]. Different types of resectoscopes, loops and fine needles were used for hysteroscopic metroplasty, whereas a 5 Fr bipolar electrode was used only in one study [21]; intrauterine device (IUD) was inserted after procedure with the aim of preventing intrauterine adhesions (IUAs) formation in three studies [18,21,22]. Double cervix was corrected in one study published in 2008: fine Metzenbaum scissors were used to perform a wedge resection of the tissue between the two cervixes and to unify the cervical ostia as well as a blind excision of the septum between the two cervixes, up to the level of the internal cervical ostium [17]. Metroplasty was performed under different guidance techniques to allow the orientation of the uterus during the procedure and to correctly incise the septum avoiding complications. A Foley catheter balloon was used in four studies [15,17,19,23] while Hegar dilators were used in three studies [16,18,24]. Patton et al. [15] used a curved plastic Pratt dilator in addition to the Foley catheter. In three studies, it was not specified whether they used guidance techniques [20,21,22]. Procedure was completed under laparoscopic guidance in six studies [15,18,19,20,22,23].

### 3.3. Fertility Outcomes

Primary fertility outcomes were pregnancy rates and miscarriages rate and are summarized in Figure 2. After surgical treatment (71/86 patients), 47 (66.2%) pregnancies were achieved: 41 (87.3%) live birth and ongoing pregnancies and six spontaneous miscarriages occurred. Preterm delivery occurred in one case. In vitro fertilization (IVF) was used in six cases to obtain conception. No malpresentations were reported. Of the 15 patients not surgically treated, eight pregnancies were achieved (53.3%): seven of them (87.5%) ended in spontaneous miscarriages and one term pregnancy was achieved (12.5%). The incidence of miscarriage was significantly lower in the treated group (Figure 2). The 15 not surgically treated patients were all described in one single study [19], which analyzed the largest sample size. It is the only article with a control group of untreated patients: 21 patients with complete uterine septum, double cervix and vaginal septum underwent surgical treatment (one was lost at follow up) and 15 women chose to remain untreated. Obstetrical outcomes were also described in this study. All deliveries were cesarean sections to avoid any risk of dystocia from the persistence of the two cervixes and the presence of vaginal scar tissue. All newborns survived. After delivery, there were two cases of placental retention in the treated group and both placentas were removed manually. No postpartum hemorrhage or uterine rupture occurred in these patients. In addition, patients reporting dyspareunia and dysmenorrhea experienced less pain after treatment. The study of Patton et al. [15] confirmed a postoperative decrease of late miscarriages, preterm births and associated neonatal mortality or morbidity. Similar findings were described in the prospective observational study of Wang et al. [18] where the miscarriage rate decreased from 100.0% (31 of 31) to 9.1% (1 of 11).

## 4. Discussion

The review showed that hysteroscopic metroplasty is an effective, safe and minimally invasive approach and should be considered the first choice for symptomatic patients affected by U2b C2 V1 malformation; it also showed that surgical treatment is associated with favorable fertility outcomes, especially in terms of the reduction in miscarriage rate and obstetrical complications, even though no reliable statistical evidence supports the intervention in asymptomatic patients in terms of fertility symptoms [25]. Surgical treatment also reduces symptomatic dyspareunia, often reported by affected patients. The lack of standardized diagnostic techniques and the low incidence of U2b C2 V1 malformation in the general population may have led to frequent misdiagnosis. Before the advent of 3D US in the diagnosis of uterine malformations, different techniques were used to diagnose this complex anomaly (TVUS, HSG, MRI). Moreover, only in 2013 a consensus paper between ESHRE and ESGE societies [24] found an agreement to correctly classify it. The triad “complete uterine septum–double cervix–longitudinal vaginal septum” can be very difficult to diagnose and many cases may frequently be misdiagnosed. A combined approach of 3D ultrasounds and diagnostic hysteroscopy [6,26] is fundamental to correctly diagnose these patients. The first step of our combined approach is the 2D-3D ultrasound that confirms the presence of a complete uterine septum and suspects two cervical canals. Diagnostic hysteroscopy is then fundamental to differentially diagnose between double cervix and single cervix with cervical septum. After entering the vagina vaginoscopically, a single cervix is visualized, with access to the first emicavity with a single tubal ostium. At this point, the presence of a vaginal septum can be suspected. Once the vaginal septum is identified, the other emivagina is entered, with the visualization of the second cervix, the other emicavity and the second tubal ostium. Lack of surgical treatment standardization may also have affected fertility outcomes. In the older studies, surgeons used less refined instruments such as Metzenbaum scissors and invasive techniques such as laparotomic assisted metroplasty, whereas minimally invasive techniques and miniaturized hysteroscopic instruments were used in the most recent studies. In the last decade, minimally invasive hysteroscopic resection became the gold standard treatment [27,28]. Hysteroscopic resection of the septum, especially when using new miniaturized devices, is a minimally invasive technique with a low complication rate and no need for extended recovery [29,30]; it also allows early conception attempts [6,31]. Moreover, a recent video article described a minimally invasive hysteroscopic resection technique using different miniaturized instruments which led to complete uterine septum and vaginal septum resection with no complications in U2b C2 V1 affected patients [32]. A 30-day post-operative office hysteroscopy control might be offered in order to evaluate the surgical result. 

### 4.1. Strengths 

Our systematic review is the first attempt to gather and summarize all the studies present in the literature concerning the diagnosis and treatment of this rare and complex Müllerian malformation and to describe the related obstetrical and fertility outcomes. The strengths of our study are the extensive search query and the quality of the studies included. In fact, we excluded publications in which the cervical assessment was not clear: we included patients with double cervix, and we excluded patients with single cervix and cervical septum. Moreover, we excluded publications concerning other types of malformations and studies that did not explain the type of treatment adopted completely and clearly. 

### 4.2. Limitations

Studies included were mainly case reports or case series, with small sample sizes, and mostly were retrospective. This is because the number of papers focusing on this rare malformation and its management is limited, mostly published before the actual ESHRE/ESGE classification system, without a clear classification of “double cervix” and septate cervix. Nevertheless, our search query was specifically oriented on “double-cervix” and each of the selected studies have been screened. The differences in classification system used between the studies also represented a limitation for analysis. Moreover, the lack of diagnostic standardized methods and of surgical techniques made statistical analysis impossible. 

## 5. Conclusions

Despite described limitations, our review allows for the conclusion that hysteroscopic treatment of this complex malformation can be safely performed, leading to improved fertility outcomes, measured as the achievement of pregnancy and a reduction in miscarriage rate. Surgical treatment also allows pain relief (dyspareunia). Further prospective high-quality studies should be performed on larger sample sizes, using integrated and standardized diagnostic methods and surgical treatment with newer miniaturized instruments, and considering patients fertility outcomes. These should provide clinicians with appropriate information for decision making, and help affected women to better understand the advantages and disadvantages of hysteroscopic surgical treatment.

## Figures and Tables

**Figure 1 jcm-12-00189-f001:**
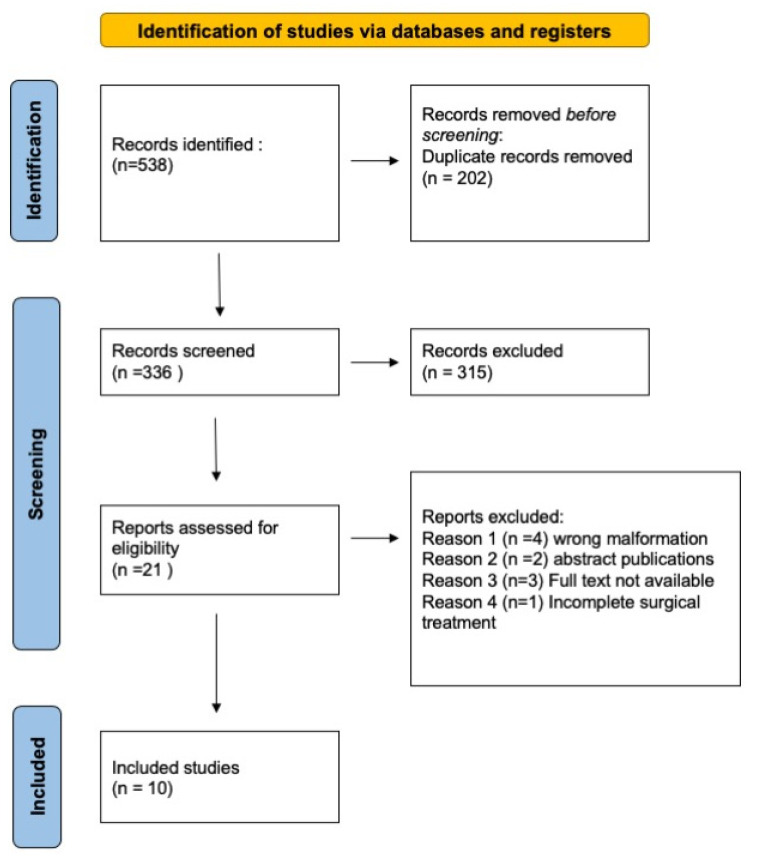
PRISMA flow diagram.

**Figure 2 jcm-12-00189-f002:**
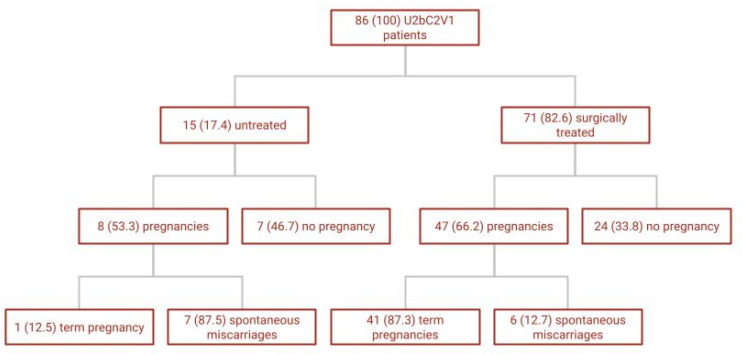
Fertility outcomes of overall included U2bC2V1 patients–N(%).

**Table 1 jcm-12-00189-t001:** Characteristics of the included studies.

Title	Author, year	Country	Study Design	Symptoms	Population Size	Method of Diagnosis	Type of Surgery	Primary Outcome	Secondary Outcome	IVF	Outcomes (Follow-Up)
*The* *diagnosis* *and* *reproductive outcome* *after* *surgical* *treatment* *of the* *complete* *septate* *uterus,* *duplicated* *cervix and* *vaginal* *septum.*	Patton et al. 2004 [15]	U.S.A.	Retrospective case series	Dyspareunia, recurrent pregnancy lost	16	TVUS, HSG, MRI	Excision of longitudinal vaginal septum. Hysteroscopic resection with operative hysteroscope and loop electrode of the uterine septum. Guidance technique: Foley balloon/ curved plastic Pratt dilator or Tompkins metroplasty (=transabdominal metroplasty).	To evaluate the reproductive outcome after metroplasty	To evaluate the best technique of metroplasty	1 patient delivered after egg donation procedures. 15 patients: not specified	12 women conceived. 17 pregnancies (14 live births or ongoing third trimester pregnancies) after hysteroscopic resection (9/12) or abdominal metroplasty (5/5); 3/17 first trimester abortions; 0/17 late abortion; 13/14 term births; 1/14 ongoing pregnancy; 0/14 preterm deliveries
*Reproductive outcome of women with rare Müllerian anomaly: report of 2 cases.*	Ignatov et al. 2008 [16]	Germany	Case report	Primary infertility Secondary infertiity	2	TVUS, Hysteroscopy	Vaginal septum resection with Metzenbaum scissors + cervical dilatation to Hegar 10 + incision of the uterine septum hysteroscopically with a loop electrod Guidance technique: Hegars dilators	Whether metroplasty increases pregnancy rate (time between procedure and conception)	Obstetric outcomes (preterm birth; vaginal delivery, CS etc.)	No	Spontaneous conception. 2 Vaginal deliveries: 38 and 35 weeks
*Leiomyoma on the septum of a septate uterus with double cervix and vaginal septum: a challenge to manage*	Caliskan et al. 2008 [17]	Turkey	Case report	Primary infertility, menorrhagia, dyspareunia	1	TVUS, HSG, MRI, Hysteroscopy	Two step surgical process: 1. Laparotomic myomectomy + excision of vaginal septum 2. Hysteroscopic metroplasty with loop electrode + resection of the tissue between the two cervixes with Metzenbaum scissors Guidance technique: pediatric Foley catheter into the new single uterine cavity inflated with 50 mL of saline solution. Antibiotic prophylaxis: Cefazolin for 5 days.	Whether metroplasty increases pregnancy rate (time between procedure and conception)	Excision of subserosal leiomyoma of the uterine septum	yes	Pregnant at 26 weeks
*Hysteroscopic septum resection of* *complete septate* *uterus with cervical* *duplication,* *sparing the double* *cervix in patients with recurrent spontaneous abortions or infertility.*	Wang et al. 2009 [18]	China	Prospective observational	Infertility, recurrent miscarriage	25	TVUS 3D-TVUS	Hysteroscopic septum resection using a 27-Fr hysteroresectoscope with a specific cutting knife electrode or cutting wire loop electrode Guidance technique: bougie as orientation and blockage of internal cervical and laparoscopic control	To evaluate intraoperative and post-operarative complications of hysteroscopic septum resection using Hank bougies (graduated metal dilators) technique	To evaluate the safety and efficacy of hysteroscopic septum resection in patients with recurrent abortions or infertility	Yes (1 pregnancy) No (14 pregnancies)	Total pregnancies. 15 Term deliveries:5; Ongoing pregnancies: 7 Abortions: 2 Preterm deliveries: 1
*Reproductive outcome following resectoscope metroplasty in women having a complete uterine septum with double cervix and vagina.*	Lin et al. 2009 [19]	China	Retrospective case-control with a concurrent control group of untreated patients	Dysmenorrhea, dyspareunia. 16 women with primary infertility; 20 women with a history of pregnancy loss	21 treated 15 untreated	3D US, HSG	Hysteroscopic resection of the uterine septum with resectoscope. Vaginal septum section with Metzenbaum scissors. Guidance technique: Foley balloon and laparoscopic control	To evaluate reproductive outcomes of women who underwent resectoscopic metroplasty: pregnancy rate, miscarriage rate, preterm delivery rate	To evaluate dyspareunia and dysmenorrhea after metroplasty, operative delivery rate, live birth rate, evaluation of adherent placentas and uterine ruptures	Not reported	Treated group: increased pregnancy rate and term delivery rate. Decreased miscarriage rate. No significant effect on primary infertility. 8/20 term deliveries. 1 patient lost to follow up 1/20 spontaneous abortion Preterm delivery: 0/9
*A mullerian anomaly ‘‘without classification’’: Septate uterus with double cervix and longitudinal vaginal septum.*	Celik et al. 2012 [20]	Turkey	Case report	Primary infertility, Dyspareunia	1	TVUS, HSG, MRI	Vaginal septum excision using scissors + hysteroscopic metroplasty (instrument not specified). Guidance technique: Laparoscopic post-procedure control	To establish the true incidence of this anomaly	\	no	Not conceived after 7 months
*Management of* *Complete vagino-* *Uterine septum in* *Patients seeking* *fertility: report of* *two cases and review* *of literature.*	Seet et al. 2015 [21]	U.K.	Case report	Primary infertility, dysmenorrh ea	2	TVUS, MRI, HSG	Resection of vaginal septum with scissors, hysteroscopic transcervical resection of the cervical and uterine septum with thin- gauge bipolar electrocautery Guidance technique: laparoscopic control	To evaluate the reproductive outcome after metroplasty	To evaluate the best technique of metroplasty. In case of cervical septum whether the cervical septum should be preserved or resected	Yes	Pt 1 not specified Pt 2 delivery at term (prophylactic cervical cerclage)
*Pregnancy after* *Hysteroscopic* *metroplasty* *under laparoscopy* *in a woman with* *complete septate* *uterus: A case report.*	Tajiri et al. 2015 [22]	Japan	Case report	Primary infertility	1	TVUS, MRI	Vaginal septectomy and hysteroscopic uterine septum resection by a loop type monopoles electrode. Guidance technique: laparoscopy	To evaluate the reproductive outcome after metroplasty	To evaluate the best technique of metroplasty	No	Spontaneous pregnancy ongoing
*Double cervix, septate uterus and longitudinal vaginal septum: a rare Müllerian anomaly with leiomyoma of the uterus*	Gezer et al. 2019 [23]	Turkey	Case report	Primary infertility, dyspareunia	1	HSG	Excision of longitudinal vaginal septum + unipolar resection of uterine septum Guidance technique: 12 F Foley catheter + laparoscopy	Whether metroplasty increases pregnancy rate (time between procedure and conception)	Excision of subserosal leiomyoma on the posterior wall of the uterine septum	Not reported	Not conceived after 6 months follow up
*A septate uterus with double cervix during two pregnancies: pregnancy outcome before and after cervix sparing metroplasty.*	Seljeflot et al. 2020 [24]	Slovenia	Case report	Primary infertility	1	3D TVUS, MRI	Surgical excision of the vaginal septum years before + dilatation of the two cervical canals with Hegar dilators+ hysteroscopic resection using resectoscope Guidance technique: Hegars dilators	Whether Metroplasty decreses the risk of preterm birth	To define the best way of diagnosis of this complex anomaly	Yes	Normal vaginal delivery at term after treatment

## Data Availability

Not applicable.

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
