# Peer review of "Complete Uterine Septum, Double Cervix and Vaginal Septum (U2b C2 V1): Hysteroscopic Management and Fertility Outcomes—A Systematic Review"

_jcm, 2022, doi:10.3390/jcm12010189_

Round 1

Reviewer 1 Report

This is a very interesting article written by Luca Parodi et al. that examines the treatment options and pregnancy outcomes for septate uterus, especially U2bC2V1 septum. It seems that there is still a lack of consensus on the treatment of septate uterus. This paper is very interesting in that it examines U2bC2V1, which is in a rather rare situation among septate uteruses. I enjoyed reading it very much. I would like you to answer a few questions and add to the text

1.   If one side of the vagina is narrow, I think the diagnosis of a vaginal septum may be difficult or missed. How was the vaginal septum diagnosed?

2.  Is it easy to classify C1 and C2? If it is partially fused, it may be difficult to distinguish them.

3.  How should we evaluate the patient after the uterine suptum is cut by hysteroscopic operation? I don't think there is any mention of intraoperative or postoperative evaluation method for the uterine cavity. I would like to see this added to the list of considerations.

Author Response

Answer to Reviewer #1.

Thank you for the comments.  We will answer to you point by point.

“This is a very interesting article written by Luca Parodi et al. that examines the treatment options and pregnancy outcomes for septate uterus, especially U2bC2V1 septum. It seems that there is still a lack of consensus on the treatment of septate uterus. This paper is very interesting in that it examines U2bC2V1, which is in a rather rare situation among septate uteruses. I enjoyed reading it very much. I would like you to answer a few questions and add to the text”

  1. If one side of the vagina is narrow, I think the diagnosis of a vaginal septum may be difficult or missed. How was the vaginal septum diagnosed?

Vaginal septum is sometimes difficult to diagnose. The diagnosis is made through clinical examination and during diagnostic hysteroscopy. It is actually easier to diagnose once the suspicious of diagnosis arise in the operator ‘s mind: after the complete septum diagnosis at ultrasound, a double cervix can be also suspected. Subsequest cervical and vagina extensive clinical examination allow the correct identification of two cervices and the vaginal septum. Hysteroscopy is fundamental in order to verify clinical suspicion.

In most cases, these patients are sent with a suspicious of general complex genital tract anomaly. In our Digital Hysteroscopic clinic we examine these patients with a combined approach: diagnostic hysteroscopy + 3D ultrasounds. The first step is 3D ultrasound that confirms the presence of a complete uterine septum and two cervical canals. Diagnostic hysteroscopy is then fundamental to differentially diagnose between double cervix and single cervix with cervical septum. You enter the vagina vaginoscopically, then you visualize one cervix, you enter the cervical canal and you access the first emicavity with a single tubal ostium. At this point, after the suspiscion of vaginal septum arised, the operator may search for it and eventually diagnoses it. Therefore, he/she can enter the other emivagina, visualize the second cervix, access the other uterine emicavity and identify the second tubal ostium. (lines 209-219 and 87-88). We removed lines 232-235 because otherwise it is repetitive.

  1. Is it easy to classify C1 and C2? If it is partially fused, it may be difficult to distinguish them.

In the previous comment, we answer also to this question. Diagnostic hysteroscopy is fundamental to differentially diagnose between double cervix and single cervix with cervical septum (lines 214-215 and 87-88)

  1. How should we evaluate the patient after the uterine septum is cut by hysteroscopic operation? I don't think there is any mention of intraoperative or postoperative evaluation method for the uterine cavity. I would like to see this added to the list of considerations.

In our institution, our patients always underwent a post operative office hysteroscopic control within 30 days from the metroplasty. We did not mention this in our paper because it’s not a description on how to correctly perform metroplasty and surgical resection of the vaginal septum, but t is a systematic review concerning the U2bC2V1 patients and the literature does not focus on the post operative control. Nevertheless, we added an insight on this in the discussion (lines 231-232)

Reviewer 2 Report

Thank you for the opportunity to review this manuscript. The topic is up-to date, clinically important and highly discussed, especially after the much recognized work of Mollo et al. Fertil Steril. 2009  (PMID: 18571168), which - notably – has been not included into the current analysis.

Minor issue: The table format is very difficult to read: please try another table orientation/ layout. Major issues: in the current year (2022) three systematic reviews with meta-analyses have been published: Jiang et al. Am J Obstet Gynecol MFM. 2022 (PMID: 36220552), Wu et al. Front Surg. 2022 (PMID: 35832500), and (in my opinion) the most informative - with a lot of subgroup analyses - of Noventa et al., J Clin Med. 2022 (PMID: 35743362). All of them related to reproductive outcomes after septum resection. All of them identified and included many more studies. All presented the results in a more differentiated and understandable way than is the case in the submitted work. The current manuscript did not even reflect the most recent publicationsthat met the inclusion criteria (e.g. Lan et al. Arch Gynecol Obstet. 2022, PMID: 36217037). I propose a thorough revision of the manuscript. This means a re-search and/or, in the case of rejection of articles included in the three concurrent systematic reviews and meta-analyses (as of 2022), a convincing justification for their non-inclusion.

Author Response

Answer to Reviewer #2.

Thank you for the revision and the comments.

“Thank you for the opportunity to review this manuscript. The topic is up-to date, clinically important and highly discussed, especially after the much recognized work of Mollo et al. Fertil Steril. 2009  (PMID: 18571168), which - notably – has been not included into the current analysis.

Minor issue: The table format is very difficult to read: please try another table orientation/ layout. Major issues: in the current year (2022) three systematic reviews with meta-analyses have been published: Jiang et al. Am J Obstet Gynecol MFM. 2022 (PMID: 36220552), Wu et al. Front Surg. 2022 (PMID: 35832500), and (in my opinion) the most informative - with a lot of subgroup analyses - of Noventa et al., J Clin Med. 2022 (PMID: 35743362). All of them related to reproductive outcomes after septum resection. All of them identified and included many more studies. All presented the results in a more differentiated and understandable way than is the case in the submitted work. The current manuscript did not even reflect the most recent publicationsthat met the inclusion criteria (e.g. Lan et al. Arch Gynecol Obstet. 2022, PMID: 36217037). I propose a thorough revision of the manuscript. This means a re-search and/or, in the case of rejection of articles included in the three concurrent systematic reviews and meta-analyses (as of 2022), a convincing justification for their non-inclusion.”

All the suggested papers are very interesting an UpToDate: management of complete uterine septum is still a very controversial topic. Unfortunately, the focus of our paper is not concerning complete uterine septum but a more complex and rarer mullerian anomaly which is the combination of complete uterine septum with double cervix and longitudinal vaginal septum (U2b C2 V1 by ESHRE classification).  Assuming this is the topic, we wanted to give a revision of literature, excluding all the papers concerning complete uterine septum alone, including all the papers you’ve suggested to add to the review.

We hope this is a convincing justification for not inclusion of those papers.

As for the table, we changed the layout as you suggested and certainly it is now more easy to read.

Thank you.

Reviewer 3 Report

I applaud your desire to present an  information about the relationship between uterine malformation U2b C2 V1 hysteroscopic metroplasty   and reproductive outcomes.

In the current literature there is a lot of papers concerning on the 

reproductive outcome after uteine malformation repair - especially uterine septum. According to the ESHRE, RCOG, ASRM, DGGG guidlines there is a lot of doubt how to manage with symptomatic women.

The presented manuscript is clear and well written. The abstract reflects the content of the article. The introduction provides a good, generalized background of the relevant research that quickly provides the reader with a good context. The objective is clearly defined.

All the methods have been conducted rigorously and  appropriate for this study.

The results are accurately described and tables where used to validate and summarize. 

The conclusions are reasonable and supported by the results.

Author Response

Answer to Reviewer #3.

“I applaud your desire to present an information about the relationship between uterine malformation U2b C2 V1 hysteroscopic metroplasty  and reproductive outcomes. In the current literature there is a lot of papers concerning on the reproductive outcome after uterine malformation repair - especially uterine septum. According to the ESHRE, RCOG, ASRM, DGGG guidlines there is a lot of doubt how to manage with symptomatic women. The presented manuscript is clear and well written. The abstract reflects the content of the article. The introduction provides a good, generalized background of the relevant research that quickly provides the reader with a good context. The objective is clearly defined. All the methods have been conducted rigorously and appropriate for this study. The results are accurately described and tables where used to validate and summarize. The conclusions are reasonable and supported by the results.”

Thank you for the revision and the positive comments; we truly appreciated them. Literature data is lacking standardization concerning diagnosis, obstetrical outcomes, and surgical treatment. The purpose of the presented paper is indeed to inspire and encourage scientific societies to propose and initiate more precise investigations upon these topics. We believe that awareness of the triade (U2b C2 V1) is the basic step in order to achieve correct diagnosis and patient-tailored treatment.

Round 2

Reviewer 2 Report

Thank you for the explanation and the corresponding clarification in the manuscript (line 87/88). I recommend accepting of the revised paper.